# Task-Oriented Training with Abdominal Drawing-in Maneuver in Sitting Position for Trunk Control, Balance, and Activities of Daily Living in Patients with Stroke: A Pilot Randomized Controlled Trial

**DOI:** 10.3390/healthcare11233092

**Published:** 2023-12-04

**Authors:** Seunghoon Jeong, Yijung Chung

**Affiliations:** 1Department of Physical Therapy, Graduate School, Sahmyook University, 815 Hwarang-ro, Nowon-gu, Seoul 01795, Republic of Korea; ptjsh211@syuin.ac.kr; 2Department of Physical Therapy, College of Health and Welfare, Sahmyook University, 815 Hwarang-ro, Nowon-gu, Seoul 01795, Republic of Korea

**Keywords:** activities of daily living, balance, stroke, task

## Abstract

In many patients with stroke, trunk stabilization and balance are decreased. Trunk stabilization and balance are important to perform posture control and functional movement. This study investigates the effects of task-orientated training with the abdominal drawing-in maneuver in a sitting position on trunk control, balance, and activities of daily living in patients with chronic stroke. The study design is a randomized controlled trial. Thirty-eight patients with chronic stroke participated in this study. The task-oriented training combined with the abdominal drawing-in maneuver group (*n* = 13), the task-oriented training group (*n* = 13), and the control group (*n* = 12) received traditional physical therapy. Three groups participated in a total of 20 sessions, four times a week for five weeks. After the five-week training period, trunk control, balance, and activities of daily living were assessed. The task-oriented training combined with the abdominal drawing-in maneuver group demonstrated significant improvements in trunk control, balance, and activities of daily living compared to the task-oriented training and control groups (*p* < 0.05). These results have implications for improving trunk control, balance, and activities of daily living for patients with stroke, and support the integration of this training method into standard clinical practice.

## 1. Introduction

Patients with stroke-induced hemiparesis experience difficulty in trunk control, balance instability, dysfunction in gait and activities of daily living (ADL) [1], and difficulty in postural control [2] because anticipatory stabilizing muscles of the trunk do not function appropriately owing to muscle weakness and sensory changes.

In patients with stroke, training in a sitting position is more unstable than that in a lying position owing to the narrow base of support. Nonetheless, it is more functional than that in a lying position and provides opportunities for greater muscle contraction to facilitate muscle activation [3]. Core stabilization exercises in the antigravity posture of sitting upright with the back straight are more effective than those in the lying position and are safe and easy to implement clinically [4,5,6,7,8]. Such exercises may improve sitting postural balance, including trunk control, and enable the manipulation of objects within arm’s reach, which is predictive of improved ADL [6,9]. Moreover, muscle activation in the paretic lower extremity is increased by facilitating weight shifting to the paretic side, and this effect is carried over with postural changes to the standing position [9,10].

Task-oriented training, one of the approaches for the rehabilitation of patients with stroke, can be used as a beneficial exercise method for those suffering from musculoskeletal and neurological disorders because it involves constructing interesting tasks to motivate patients and causes neural reorganization during the repetitive training process [11]. Task-oriented training is also a neurological therapeutic approach that improves walking ability and the quality of life by improving motor control and balance [9,12,13]. This training can improve performance of ADL based on the activation of functional movement via motor learning [14,15,16].

The purpose of trunk stabilization exercises is to strengthen the deep muscles, promote synergistic effects, and improve trunk stability [17]. Of the various trunk stabilization exercises, the abdominal drawing-in maneuver (ADIM) has been proven to be more effective in increasing the thickness of the transverse abdominis than typical core exercises [17,18,19]. ADIM draws the lower abdomen in without spinal or pelvic movement to increase intra-abdominal pressure (IAP) via preferential contraction of the transverse abdominis and internal obliques, thus reducing excessive spinal anterior flexion and an anterior pelvic tilt [20,21,22,23,24,25,26]. To accurately perform ADIM, a pressure biofeedback unit (PBU), a tool developed to help retrain muscle activities, is available. This tool can provide useful visual biofeedback during treatment [2,25,27,28,29,30]. Visual feedback-based training can be beneficial in improving balance and stimulating interest in stroke rehabilitation [8]. During visual feedback, as we observe our actions, mirror neurons are activated, leading to the formation of new neural pathways in the primary motor cortex when imitating movements [9]. Therefore, providing information through visual feedback while performing actions can be helpful for patients [8].

To date, some studies have reported positive results for trunk control, balance, and gait when applying a single exercise with ADIM to patients with stroke. However, most of these exercises were performed in the supine or prone position, and there is a lack of research on ADIM in the sitting position for functional activities. Therefore, this study aimed to investigate the effects of task-oriented training combined with ADIM in a sitting position on trunk control, balance, and ADL in patients with stroke, and to compare task-oriented training in a sitting position and its effectiveness.

## 2. Materials and Methods

### 2.1. Study Design and Ethical Considerations

This study was a randomized controlled trial. The groups were categorized as the task-oriented training combined with ADIM group, the task-oriented training group without ADIM group, and the control group according to the treatment method. Pre-training and post-training surveys were conducted. Before the experiment, the participants signed a voluntary consent form for this study, and the study was approved by the Institutional Review Board of Sahmyook University (approval number: 2-7001793-AB-N-012018053HR). Further, this trial has been registered on the clinical trials register (approval number: KCT0008750) and has been designed and reported in accordance with the CONSORT guidelines. The study was conducted according to the guidelines of the Declaration of Helsinki.

### 2.2. Participants

Forty-five patients with stroke who were hospitalized at Bobath Memorial Hospital, Gyeonggi-do, Republic of Korea, were recruited for this study. The following patients were included in the study: (1) those who had been diagnosed with hemiparesis owing to stroke by a neurologist and rehabilitation specialist; (2) those who developed a brain injury > 6 months ago [27]; (3) those who had a Mini Mental State Examination-Korean (MMSE-K) score of ≥21, with the ability to communicate and follow instructions [31,32]; (4) those who had a Trunk Impairment Scale (TIS) score of ≥10 [4]; (5) those who did not have any orthopedic problems that may affect their sitting balance [4]; and (6) those could perform reaching, grasping, and kicking tasks. Patients with difficulty in understanding the study and those with vestibular or cerebellar disease were excluded, and all participants were screened using the selection and exclusion criteria, resulting in 38 participants. Patients were enrolled between 6 August 2018 and 28 September 2018.

All participants received a detailed explanation of the experimental procedure and completed an informed consent form. Using G*Power Version 3.1.9.4 (Heinrich-Heine-University Düsseldorf, version 3.1.9.4, Düsseldorf, Germany) [33], and applying an effect size 0.25, 1-beta error 0.80, and alpha error 0.05, a sample size of 36 patients was required, and 45 patients were selected considering the dropout rate. A total of 38 patients participated. The selected participants were randomly www.randomizer.org (accessed on 30 May 2018) assigned to the task-oriented training combined with ADIM group (*n* = 13), the task-oriented training group (*n* = 13), and the control group (*n* = 12) via lottery (Table 1).

### 2.3. Equipment and Data Collection

Trunk control in all study participants was measured using the TIS and the Postural Assessment Scale for Stroke (PASS) before and after the 5-week experiment. Balance ability was assessed using the Functional Reaching Test (FRT), Berg Balance Scale (BBS), and the Messen–Trainieren–Dokumentieren (MTD) system. Additionally, ADLs were measured and evaluated using the Modified Barthel Index (MBI).

All participants performed the given exercises for 30 min a day, 4 days a week for 5 weeks, for a total of 20 sessions. All three groups also received conventional physical therapy for 30 min per session, twice a day, 5 days a week for 5 weeks according to the hospital inpatient rehabilitation program.

### 2.4. Procedures and Interventions

The task-oriented training combined with ADIM group performed task-oriented training while doing ADIM using a PBU in a sitting position. The task-oriented training group performed task-oriented training only in a sitting position, and the control group performed normal exercises. The task-oriented training comprised arm extension, ring stacking, and ball kicking, and each exercise was performed 15 times per set, with 3-set repetitions.

#### 2.4.1. Task-Oriented Training Combined with ADIM

In task-oriented training combined with ADIM, a PBU (Chattanooga Stabilizer, Chattanooga Group Inc., Hamilton Country, Tennessee, USA) was used to provide visual feedback of exercise during ADIM. It was performed in a sitting position at a table with a straight back, both feet on the floor, and knee joint and hip joint angles maintained at 90 degrees. The participant was instructed by the researcher to slowly draw the lower abdomen toward the umbilicus, with no movement of the upper abdomen or spine between movements, and the pelvis maintained in a neutral position [27]. The participants were asked to perform the maximal ADIM in a sitting position, and a PBU calibrated to 70 mmHg was placed at the anterior part of the abdomen, secured with a strap (Figure 1a). The increased pressure reading when the contractile force of the ADIM was removed was measured and recorded, and the expansion to 80 mmHg is one example of an increase when ADIM contraction force is removed (Figure 2b). Values measured during ADIM with the pressure on the PBU placed at the anterior part of the abdomen being set to 70 mmHg were compared with the previously measured values. If a reading of ≥80% was maintained, ADIM was construed as successful and the training proceeded. (Figure 1c). Therefore, visual biofeedback was used while the patients were looking at the pressure gauge so that they could maintain the ADIM during the task-oriented training, and when the pressure reading dropped, a therapist provided auditory biofeedback [13,34,35] (Figure 1). Minimal assistance was provided by the therapist if the patient had difficulty performing the task independently. A 5 min warm-up was performed before and after the start of task-oriented training [36]. Additionally, to maintain consistency in the intensity of the ADIM, the maximum ADIM values were measured repeatedly while the patients performed the first exercise of each week and the changes in values were applied to the exercises (Figure 2).

#### 2.4.2. Task-Oriented Training

Task-oriented training was performed in a sitting position with a straight back, both feet on the floor, and knee joint and hip joint angles at 90 degrees. Exercises of arm reaching the limit of stability, ring stacking, and ball kicking were performed 15 times in one set with 3-set repetitions for each exercise [13,34,35] without PBU, with a 5 min warm-up before and after the task-oriented training [36] (Figure 2).

#### 2.4.3. General Physical Therapy

Of the neuro-facilitation approaches based on the conventional motor development theory and motor learning theory, rolling, sit-to-stand, stand-to-sit, and maintaining stable standing treatment were provided for general physical therapy. This approach facilitated movement and aided in performing various functions by re-educating patients with stroke on abnormal movements and postures, with treatment to help them in their daily life [37,38,39]. All participants received 30 min of general physical therapy twice a day, 5 days a week for 5 weeks, and additionally, the control group received the same amount of general physical therapy as the experimental group.

### 2.5. Measurement of the Results

In this study, the general characteristics of the participants were measured before the assessment. The results were measured using clinical tests, with high reliability, validity, and sensitivity as dependent variables. The TIS and PASS were used to assess trunk control; the FRT, BBS, and weight bearing were used to assess balance; and the MBI was used to measure ADL.

TIS is a 17-item instrument that assesses static and dynamic control and coordination of the trunk in a sitting position, comprising a total score of 23 points. The test–retest reliability in patients with stroke was high, with an ICC of 0.92 [1]. PASS is a 12-item scale with a minimum score of 0 and a maximum score of 3 for each item, for a total score of 36 points. The Korean version of tmhe PASS used in this study has a high reliability of 0.97 [39].

FRT tests dynamic balance and uses the average of three measurements as the result. The test method is easy and convenient to apply in clinical settings, with an intra-rater reliability of *r* = 0.98 and inter-rater reliability of *r* = 0.99 [40]. BBS for measuring balance comprises 14 items and is a highly valid assessment tool with an internal consistency of 0.97 in patients with stroke [41].

To measure weight bearing, the Messen–Trainerieren–Dokumentieren (MTD, balance version 4.0) system was used, which can measure two-footed balance ability and left–right weight bearing in a standing position and when rising from a sitting position [42]. MBI comprises 10 items, with a minimum score of 0 and a maximum score of 100, with higher scores indicating greater independence in daily living. The inter-rater reliability of the MBI is *r* = 0.95 and the intra-rater reliability is *r* = 0.89 [43]. The Korean version of the BBS used in this study has a very high reliability of 0.98 [44].

### 2.6. Statistics

This study used the SPSS version 21.0 (SPSS Inc., Chicago, IL, USA) program for analysis. Normality was tested using the Shapiro–Wilk test. Participants were tested for homogeneity in general characteristics and dependent variables before the experiment. Pre- and post-experimental changes within the groups were compared using paired sample *t*-test, differences in post-experimental changes between groups were compared using one-way analysis of variance, and post hoc tests were performed using Scheffe. The statistical significance level for all data was <0.05.

## 3. Results

### 3.1. Trunk Control

The TIS and PASS scores were significantly improved in the task-oriented training combined with ADIM group compared with the task-oriented training group and the control group (*p* < 0.05). Additionally, the task-oriented training group showed significant improvement in TIS and PASS scores compared with the control group (*p* < 0.05) (Figure 3a,b and Table 2). 

### 3.2. Balance

The FRT, BBS, and weight bearing ratio values were significantly improved in the task-oriented training combined with ADIM group compared with the task-oriented training and control groups (*p* < 0.05). Additionally, the task-oriented training group showed significant improvement in FRT, BBS, and weight bearing ratio values compared with the control group (*p* < 0.05) (Figure 3c–e and Table 3).

### 3.3. ADL

The MBI scores used to measure ADL were significantly improved in the task-oriented training combined with ADIM group compared with the task-oriented training group and the control group (*p* < 0.05) (Figure 3f and Table 4).

## 4. Discussion

This study examined the effects of task-oriented training combined with ADIM in a sitting position on trunk control, balance, and ADL in patients with stroke. The results demonstrated that the training was effective in improving trunk control, balance, and ADL.

The improvement in trunk control with the task-oriented training combined with ADIM in the present study could be attributed to the following reasons. The trunk muscles were used as primary or cooperative muscles in voluntary trunk control, providing an automatic response to unexpected body shaking [2,45,46]. Vasseljen and Fladmark [47] reported that the thickness of the transverse abdominis increased by 3% in patients with low back pain when real-time ultrasound imaging was applied along with ADIM. In addition, Seo et al. [48] applied trunk stabilization exercises using a PBU and ADIM in patients with stroke, resulting in a significant increase in the thickness of transverse abdominis (*p* < 0.05). The ADIM, which was performed using a PBU for voluntary stabilization and control of the trunk, allowed for appropriate simultaneous contraction of the transverse abdominis, internal obliques, and erector spinae muscles [20,49,50]. In a study by Haruyama et al. [27], 32 patients with stroke underwent core stability training with ADIM, and there was a significant difference in TIS scores, from 15.50 ± 3.33 points before the experiment to 19.63 ± 2.45 points after the experiment (*p* < 0.05). Cabanas-Valdes et al. [4] divided 80 patients with stroke into two groups to improve trunk control, dynamic sitting balance, and standing balance by applying core stability exercises. The mean (±SD) differences between the groups’ total TIS, BBS, and Barthel Index scores were 3.40 ± 4.12 (*p* < 0.05), 14.54 ± 18.19 (*p* < 0.05), and 13.17 ± 25.27 (*p* < 0.05) points, respectively. Furthermore, providing visual and auditory biofeedback improved trunk control by sustaining the exact contractile force of the trunk muscle, allowing them to experience contractions in different environments. While maintaining this condition, task-oriented training, including arm extension movements, was performed, which is deemed to be the reason for improved trunk control [51]. Task-oriented training combined with ADIM could be an effective treatment method for patients with stroke.

In this study, the improvement in balance ability achieved with the task-oriented training combined with ADIM could be ascribed to the fact that improved trunk control provided a stable foundation for postural control, thus enabling upright posture to perform selective trunk movements and weight shifts [1,52]. Jung et al. [51] conducted a study involving training using ADIM along with visual biofeedback, and found that changes in static sitting balance ability, medial–lateral sway speed, anterior–posterior sway speed, and velocity of moment in both the experimental and control groups displayed significant improvement after the intervention, regardless of their vision. Additionally, the improvement was more significant in the experimental group than in the control group (*p* < 0.05). In Haruyama’s study [27], ADIM training enhanced the trunk control ability of patients with stroke (*p* < 0.05), and the Brief-BESTest balance test scale score was 5.43 ± 3.47 points in the experimental group before the experiment. The results showed a significant difference in the control group, with 8.22 ± 4.67 points after the experiment (*p* < 0.05).

Furthermore, trunk stabilization training in a sitting position improved seated dynamic postural balance, upright balance, gait, and ADL in patients with stroke [4]. Moreover, arm extension exercise during ADIM caused an increase in the thickness of transverse abdominis [53]. This muscle is the first to be activated among the abdominal muscles before limb movements occur and is responsible for generating and regulating IAP and increasing lumbar spine stability to allow for trunk control [53,54]. The increased thickness of the transverse abdominis muscle contributed to trunk stability and balance [11,55]. These elements provided a stable foundation for balance and ADL via the integrated coordination of fine postural control and motor performance of the patient’s trunk [52]. This promoted stability and facilitated movements, which might have aided in the recovery of balance ability.

The results of this study showed that ADLs were improved significantly based on the pre–post difference in the group of task-oriented training with ADIM in a sitting position compared with the task-oriented training and control groups. This is because the ADIM, commonly used clinically for trunk control, stabilizes the trunk via precise and selective contraction of the transverse abdominis, internal obliques, and external oblique abdominis, located deeper than the superficial abdominal muscles [22,56]. ADIM promotes neuromuscular re-education by inducing contraction of the transverse abdominis and internal obliques to enhance lumbar spine stability [22,25,56], resulting in improved trunk control and balance.

Additionally, more studies have reported a significant correlation in validity between the BBS and the MBI, which could be attributed to trunk control and balance. In a study by Cabanas-Valdes et al. [4], significant differences were observed in the scores of both the TIS and BBS in patients with chronic stroke (*p* < 0.05). Additionally, these findings had a significant impact on the ADL measurement tool, the Barthel index, with scores increasing from 32.00 ± 15.27 points before the intervention to 68.50 ± 22.37 points after the intervention (*p* < 0.05). In a study by Karaca et al. [1] involving 36 patients with chronic stroke investigating the effect of trunk, upper limb, and lower limb functions on ADL and balance, the multiple regression analysis revealed that the TIS scores had a positive influence on both the BBS and Barthel index scores (*p* < 0.05). Furthermore, improvements in lower limb function were found to have a significant impact on BBS scores (*p* < 0.05).

Trunk control and balance are highly correlated and mutually influential to ADL [52,57]. This study found that task-oriented training combined with ADIM improved ADL in patients with stroke by augmenting their trunk control and balance.

In general, the ultimate rehabilitation goal for patients with stroke is to perform ADL independently [58]. This study investigated the effects of task-oriented training combined with ADIM in a sitting position on trunk control, balance, and ADL in patients with stroke. This approach was found to be effective in restoring trunk control and balance, as well as facilitating the recovery of ADL in the patients.

The limitations of this study include the recruitment of inpatients from a specific region and the utilization of a small sample size, potentially limiting its generalizability. Further studies are needed with larger sample sizes to investigate the impact of interventions in more functional positions than sitting on the rehabilitation of patients with stroke.

## 5. Conclusions

These findings demonstrate that task-oriented training with the use of PBU for ADIM application in the TASK + ADIM group contributed to increased trunk control, balance, and improvements in ADL for patients with stroke. TASK + ADIM was especially superior to TASK for ADL. We advocate for subsequent longitudinal research encompassing broader and more varied cohorts of patients with stroke to both validate and enrich our findings on this critical subject.

## Figures and Tables

**Figure 1 healthcare-11-03092-f001:**
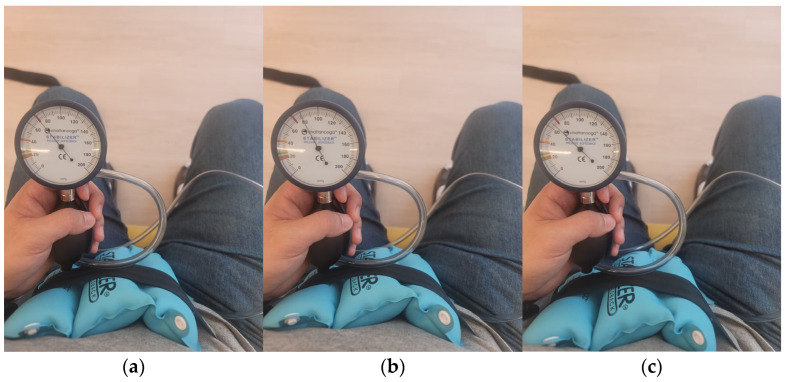
Method of using PBU: (**a**) perform a maximal ADIM using a strap to set the PBU pressure to 70 mmHg; (**b**) check the increase in PBU pressure when maximal ADIM is reduced; (**c**) perform the task while maintaining ADIM at 80% of (**a**,**b**).

**Figure 2 healthcare-11-03092-f002:**
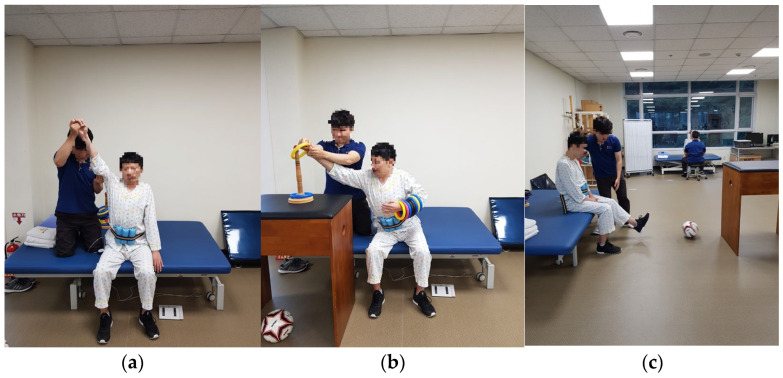
Task-oriented training with ADIM: (**a**) reaching the limit of stability; (**b**) hooking with a ring; (**c**) ball kicking.

**Figure 3 healthcare-11-03092-f003:**
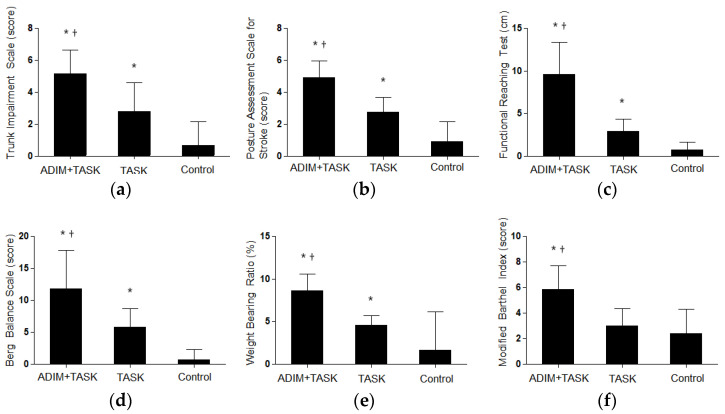
Comparison of change values in trunk control, balance, and activities of daily living in each group: (**a**,**b**) trunk control; (**c**–**e**) balance; (**f**) activities of daily living. * Significant improvement from the control group; ^†^ Significant improvement from the TASK group.

**Table 1 healthcare-11-03092-t001:** General and medical characteristics of the subjects (*n* = 38).

Variable	TASK + ADIM (*n* = 13)	TASK (*n* = 13)	Control (*n* = 12)	*F/X*^2^ (*p*)
Age (years)	51.08 ± 11.84 *	51.08 ± 11.84	48.33 ± 10.81	0.249 (0.781) ^a^
Body mass index (kg/cm^2^)	23.37 ± 2.90	22.47 ± 2.30	22.71 ± 2.83	0.390 (0.680) ^a^
Gender (male/female)	7/6	8/5	7/5	0.159 (0.924) ^b^
Etiology (infarction/hemorrhage)	6/7	6/7	7/5	1.000 (0.607) ^b^
Paretic side (right/left)	6/7	7/6	7/5	0.698 (0.709) ^b^
Post-stroke duration (month)	15.85 ± 5.74	13.31 ± 4.96	14.42 ± 5.43	0.726 (0.491) ^a^
Korean-Mini Mental State Examination (score)	26.54 ± 2.96	25.69 ± 2.29	25.50 ± 2.61	0.559 (0.577) ^a^
Modified Ashworth Scale—elbow joint (score)	1.04 ± 0.96	1.11 ± 0.82	0.79 ± 0.062	2.901 (0.821) ^b^

Note: TASK + ADIM: task-oriented training combined with ADIM; TASK: task-oriented training without ADIM. * Mean ± SD; ^a^ One-way ANOVA; ^b^ Chi-square test.

**Table 2 healthcare-11-03092-t002:** Comparison of trunk control in each group (*n* = 38).

Variable	TASK + ADIM (*n* = 13)	TASK (*n* = 13)	Control (*n* = 12)	*F* (*p*)
Trunk impairment scale (score)				
Before training	13.77 ± 2.24 ^a^	13.46 ± 2.37	13.75 ± 1.86	0.081 (0.923)
After training	18.92 ± 1.55	16.23 ± 1.69	14.42 ± 2.07	
Change values	5.15 ± 1.46 *^†^	2.77 ± 0.51 *	0.67 ± 1.50	24.338 (<0.001)
*t* (*p*)	−12.700 (<0.05)	−5.448 (<0.05)	−1.542 (<0.05)	
Posture assessment scale of stroke (score)				
Before training	29.92 ± 1.50	29.77 ± 2.35	29.67 ± 2.31	0.048 (0.953)
After training	34.85 ± 1.21	32.54 ± 1.94	30.58 ± 2.43	
Change values	4.92 ± 1.04 *^†^	2.77 ± 0.93 *	0.92 ± 1.24	43.858 (<0.001)
*t* (*p*)	−17.105 (<0.05)	−10.773 (<0.05)	−2.561 (<0.05)	

Note: TASK + ADIM: task-oriented training combined with ADIM; TASK: task-oriented training without ADIM. ^a^ Mean ± SD; * Significant improvement from the control group; ^†^ Significant improvement from the TASK group.

**Table 3 healthcare-11-03092-t003:** Comparison of balance in each group (*n* = 38).

Variable	TASK + ADIM (*n* = 13)	TASK (*n* = 13)	Control (*n* = 12)	*F* (*p*)
Functional reaching test (cm)				
Before training	20.64 ± 5.63 ^a^	23.28 ± 5.32	20.86 ± 4.79	0.994 (0.380)
After training	30.28 ± 4.67	26.21 ± 4.62	21.64 ± 4.92	
Change values	9.64 ± 3.75 *^†^	2.92 ± 1.47 *	0.78 ± 0.91	46.509 (<0.001)
*t* (*p*)	−9.263 (<0.05)	−7.186 (<0.05)	−2.948 (<0.05)	
Berg balance scale (score)				
Before training	39.23 ± 8.57	39.15 ± 8.96	37.75 ± 8.87	0.110 (0.896)
After training	51.08 ± 3.45	45.00 ± 7.81	38.50 ± 7.85	
Change values	11.85 ± 5.94 *^†^	5.85 ± 2.88 *	0.75 ± 1.60	24.497 (<0.001)
*t* (*p*)	−7.188 (<0.05)	−7.313 (<0.05)	−1.621 (<0.05)	
Weight bearing ratio (%)				
Before training	41.62 ± 1.50	41.92 ± 2.29	41.92 ± 2.31	0.093 (0.911)
After training	50.23 ± 2.42	46.54 ± 1.80	43.58 ± 5.45	
Change values	8.62 ± 1.94 *^†^	4.62 ± 1.12 *	1.67 ± 4.48	19.011 (<0.001)
*t* (*p*)	−16.027 (<0.05)	−14.846 (<0.05)	−1.289 (<0.05)	

Note: TASK + ADIM: task-oriented training combined with ADIM; TASK: task-oriented training without ADIM. ^a^ Mean ± SD; * Significant improvement from the control group; ^†^ Significant improvement from the TASK group.

**Table 4 healthcare-11-03092-t004:** Comparison of activities of daily living in each group (*n* = 38).

Variable	TASK + ADIM (*n* = 13)	TASK (*n* = 13)	Control (*n* = 12)	*F* (*p*)
Modified Barthel Index (score)				
Before training	67.15 ± 6.72 ^a^	67.62 ± 7.09	68.17 ± 8.00	0.061 (0.941)
After training	73.00 ± 6.93	70.62 ± 6.38	71.25 ± 11.08	
Change values	5.85 ± 1.86 *^†^	3.00 ± 1.35	2.42 ± 1.88	14.617 (<0.001)
*t* (*p*)	−11.308 (<0.05)	−7.989 (<0.05)	−4.451 (<0.05)	

Note: TASK + ADIM: task-oriented training combined with ADIM; TASK: task-oriented training without ADIM. ^a^ Mean ± SD; * Significant improvement from the control group; ^†^ Significant improvement from the TASK group.

## Data Availability

The data that support the findings of this study are available from the corresponding author upon reasonable request.

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
