# Peer review of "Task-Oriented Training with Abdominal Drawing-in Maneuver in Sitting Position for Trunk Control, Balance, and Activities of Daily Living in Patients with Stroke: A Pilot Randomized Controlled Trial"

_healthcare, 2023, doi:10.3390/healthcare11233092_

Round 1

Reviewer 1 Report (New Reviewer)

Comments and Suggestions for Authors

Author Response

Reviewer 2 Report (New Reviewer)

Comments and Suggestions for Authors

Thank you for submitting your valuable manuscript to this journal.

This is a randomized controlled trial on 38 participants with hemiparesis after stroke. In this study the additive effects of abdominal drawing-in maneuver (ADIM) to task-oriented training in sitting position on trunk control, balance, and activities of daily living (ADL) are investigated and compared to the control group with routine treatment protocol. Participants were randomly assigned to the task-oriented training combined with ADIM group, the task-oriented training group and the control group.

This is an interesting intervention on stroke patients, but there are some points that need to be considered in the manuscript:

1-      Why did you recruit the hospitalized patients? Since their brain injury was more than 6 months ago, why were they hospitalized?

2-      Trunk Impairment Scale (TIS) score of the patients in different groups are missing in table 1.

3-      What is modified Ashworth Scale (MAS) score in table 1?

4-      The abdominal drawing-in maneuver (ADIM) using pressure biofeedback unit (PBU) is not clearly explained in section 2.4.1. Inflating the abdominal PBU to 80 mmHg is not mentioned in the text and it is just on the figure. A successful ADIM was maintaining the 80% of which pressure? (70 or 80 mmHg) Please explain the PBU process in the text more clearly.

5-      Please revise the conclusion of the study and also the conclusion in the abstract. Indeed the results of this study showed some additive effects for TASK+ADIM and TASK to general physical therapy program on trunk control and balance in the included participants. TASK+ADIM was superior to Task on ADL.

6-      You may add the sitting “position” to the title of the manuscript.

Good luck.

Comments on the Quality of English Language

I highly recommend English editing for this manuscript.

Author Response

Reviewer 3 Report (New Reviewer)

Comments and Suggestions for Authors

First of all, the uploaded version of the manuscript is not a final version, it is with modifications highlighted with change control. This denotes sloppiness on the part of the authors. I don't know if it is worth continuing with the revision without knowing if it is the correct version.

Revise all the abbreviation in the manuscript, as an example, TASK appears sometimes abbreviated, others not…

- Comments on the abstract:

Avoid the use of abbreviations in the abstract.

- Comments on the introduction:

Line 31. Missed comma. “Disfunction in gait, and activities of daily living. Check all the manuscript as there are some missing commas in other sections.

Line 53. Missed space before the reference.

- Comments on material and methods

Check all commas and semicolons in the footnotes of all tables.

People that appear in figure must be censored.

The significant must be set at <0.05, not <=0.05.

Round 2

Reviewer 3 Report (New Reviewer)

Comments and Suggestions for Authors

There are still abbreviations in the abstract

Author Response

This manuscript is a resubmission of an earlier submission. The following is a list of the peer review reports and author responses from that submission.

Round 1

Reviewer 1 Report

Comments and Suggestions for Authors

1)     “People-first” language is preferred. An example is a preference for “patients with stroke” rather than “stroke patients.”

2)     Was a power (sample-size) analysis conducted before initiating the project?

3)     I prefer that authors label the first column of tables.

4)     The authors indicate in Table 1 that F statistics were used to determine differences. I doubt such is the case for gender, etiology, paretic side or MAS score.

5)     I like how the authors measured “drawing in.” However, an illustration would be helpful.

6)     Barthel should be capitalized.

7)     I’m not sure why “Parkinson’s disease was listed as a key word.

Comments on the Quality of English Language

See elsewhere

Author Response

Thank you for your generous suggestion. You have carefully pointed out the errors we have made in writing the study. Your suggestion makes us more scientific view. We expect our attention to these aspects will contribute to the clarity and rigor of our study. If there is anything missing in our thesis, please don’t hesitate to contact us. We will promptly address and fix any issues.

Comment #1

“People-first” language is preferred. An example is a preference for “patients with stroke” rather than “stroke patients.”

Response: According your comment, we corrected words. Please see the revised manuscript.

Comment #2

Was a power (sample-size) analysis conducted before initiating the project?

Response: We added sample-size analysis in method.

Comment #3

I prefer that authors label the first column of tables.

Response: We labeled first column of tables in manuscript.

Comment #4

The authors indicate in Table 1 that F statistics were used to determine differences. I doubt such is the case for gender, etiology, paretic side or MAS score.

Response: You are correct. According to your comment, we corrected the gender, etiology, paretic side or MAS score. Please see the revised manuscript.

Comment #5

I like how the authors measured “drawing in.” However, an illustration would be helpful.

Response: Good suggestion. We reflected as a Figure 1.

Comment #6

Barthel should be capitalized.

Response: According to your comment, We corrected.

Comment #7

I’m not sure why “Parkinson’s disease was listed as a key word.

Response: This is not confirmed in the manuscript. If it was reflected when uploading to the site, I will delete it.

Reviewer 2 Report

Comments and Suggestions for Authors

First of all, congratulations to the researchers for their efforts.

I will now proceed with the review of the submitted article:

Introduction: It should be improved, especially the references, as they are not up to date.

Study design and ethics committee:

How the randomisation was done (it should be explained, following the CONSORT methodology), a clinical trial should be presented to know when the patients were recruited and the study was completed.

Regarding the inclusion criteria for the task-oriented exercises, it is not specified that the patient needs an optimal upper limb to be able to perform them (e.g. reaching with hoops), as for example a patient with hemiparesis with a spatial flexor pattern would find it difficult.

As far as general physiotherapy is concerned, from my point of view (as a Bobath therapist) it is not correct to talk in a scientific article about a general therapy and explain that it has neuro-developmental approaches, etc... as this concept focuses on the individual and the same therapy cannot be applied to all patients in the same way, so I recommend calling it neurorehabilitation therapy (to avoid this mistake) and explaining which type of exercise has been applied to all in the same way.

Discussion.

The importance of central stability (core stability) is mentioned, especially in the abdominal area, but the exercises presented work laterally, i.e. gluteal stability.

As far as the tool used is concerned, the specific concentric contraction is important, but the eccentric one is not important ??

This discussion needs to be further refined, together with the references used.

Author Response

Thank you for your generous suggestion. You have carefully pointed out the errors we have made in writing the study. Your suggestion makes us more scientific view. We expect our attention to these aspects will contribute to the clarity and rigor of our study. If there is anything missing in our thesis, please don’t hesitate to contact us. We will promptly address and fix any issues.

Comment #1

Introduction: It should be improved, especially the references, as they are not up to date.

Response: Thank you for your comment. We updated the references in Introduction.

Comment #2

Study design and ethics committee: How the randomization was done (it should be explained, following the CONSORT methodology), a clinical trial should be presented to know when the patients were recruited and the study was completed.

Response: Thank you for your comment. We explained that following the CONSORT methodology, we presented how to recruited. Please see the manuscript.

Comment #3

Study design and ethics committee: Regarding the inclusion criteria for the task-oriented exercises, it is not specified that the patient needs an optimal upper limb to be able to perform them (e.g. reaching with hoops), as for example a patient with hemiparesis with a spatial flexor pattern would find it difficult.

Response: Good suggestion. We added the inclusion criteria the specific explanation. Please see the manuscript.

Comment #4

  Study design and ethics committee : As far as general physiotherapy is concerned, from my point of view (as a Bobath therapist) it is not correct to talk in a scientific article about a general therapy and explain that it has neuro-developmental approaches, etc... as this concept focuses on the individual and the same therapy cannot be applied to all patients in the same way, so I recommend calling it neurorehabilitation therapy (to avoid this mistake) and explaining which type of exercise has been applied to all in the same way.

Response: Thank you for your comment. According to your comment, we changed the general physiotherapy to neurorehabilitation therapy and added the explaining.

Comment #5

Discussion : The importance of central stability (core stability) is mentioned, especially in the abdominal area, but the exercises presented work laterally, i.e. gluteal stability.

Response: Good suggestion. We added evidence from studies showing that the core stability is also effective for function. Please see the manuscript.

Comment #6

Discussion : As far as the tool used is concerned, the specific concentric contraction is important, but the eccentric one is not important ??

Response: Good suggestion. In previous studies, ADIM is a method of inducing concentric contraction. Later we will consider studies that effect of eccentric contraction.

Comment #7

Discussion : This discussion needs to be further refined, together with the references used.

Response: According to your suggestion, we refined. Please see the manuscript.

Reviewer 3 Report

Comments and Suggestions for Authors

I recommend that the study be updated and submitted to other journals with a lower impact factor, where this pilot study can be included.

Although the study is interesting for the topic it addresses, I am mainly concerned about the small sample size to be able to extrapolate the results to the scientific community, which is a rather important limitation.

Title: due to the small sample size, I suggest putting pilot study.

Introduction:

It would be convenient to restructure it, especially in the background and justification of the study.

methodology

The study is not registered on clinicaltrials.gov.

The study is conducted in a specific region or area, those patients may have certain characteristics, so it would be difficult to generalise the results. Although they have specified limitations, this together with the small sample size makes it difficult to generalise the results.

Discussion

A restructuring would be necessary, with emphasis on the discussion of the results obtained with previously published studies.

References

Many of the references do not follow the standards of the journal.

The references need to be updated, some of them are outdated.

Author Response

Thank you for your generous suggestion. You have carefully pointed out the errors we have made in writing the study. Your suggestion makes us more scientific view. We expect our attention to these aspects will contribute to the clarity and rigor of our study. If there is anything missing in our thesis, please don’t hesitate to contact us. We will promptly address and fix any issues.

Comment #1

Title: due to the small sample size, I suggest putting pilot study.

Response: According to your comment, we added “pilot study” in end of title. Please see the revised title.

Comment #2

Introduction : It would be convenient to restructure it, especially in the background and justification of the study.

Response: Thank you for your comment. We restructured the references in Introduction.

Comment #3

Method : The study is not registered on clinicaltrials.gov.

Response: According to your suggestion, We added on CRIS(Clinical Research information Service, CRIS). Please see the method part in manuscript.

Comment #4

  Method : The study is conducted in a specific region or area, those patients may have certain characteristics, so it would be difficult to generalise the results. Although they have specified limitations, this together with the small sample size makes it difficult to generalise the results.

Response: Good suggestion. We added two difficulties were described in terms of limitations part in discussion .

Comment #5

Discussion : A restructuring would be necessary, with emphasis on the discussion of the results obtained with previously published studies.

Response: According to your suggestion, we refined. Please see the manuscript.

Comment #6

Reference : Many of the references do not follow the standards of the journal.

Response: Thank you for your comment. We modified it by referring to the journal guide line. Please see the discussion part in manuscript.

Comment #7

Reference : The references need to be updated, some of them are outdated.

Response: You are correct. According to your comment, We corrected. Please see the manuscript.

Round 2

Reviewer 2 Report

Comments and Suggestions for Authors

I congratulate the authors for their efforts and for taking the reviewers' comments into account.

I wish them well.

Reviewer 3 Report

Comments and Suggestions for Authors

Although the study is interesting for the topic it addresses, I am mainly concerned about the small sample size to be able to extrapolate the results to the scientific community, which is a rather important limitation